# The Impact of Organised Sport, Physical Education and Active Commuting on Physical Activity in a Sample of New Zealand Adolescent Females

**DOI:** 10.3390/ijerph18158077

**Published:** 2021-07-30

**Authors:** Jennifer T. Gale, Jillian J. Haszard, Tessa Scott, Meredith C. Peddie

**Affiliations:** 1Department of Human Nutrition, University of Otago, Level 7, Science 2 Building, 70 Union Street, Dunedin 9016, New Zealand; tessa.scott@otago.ac.nz (T.S.); meredith.peddie@otago.ac.nz (M.C.P.); 2Division of Sciences, University of Otago, 85 Union Place West, Dunedin 9016, New Zealand; jill.haszard@otago.ac.nz

**Keywords:** physical activity, active commuting, organised sport, physical education

## Abstract

Background: The majority of adolescents do less physical activity than is recommended by the World Health Organization. Active commuting and participation in organised sport and/or physical education individually have been shown to increase physical activity in adolescents. However, how these domains impact physical activity both individually and in combination has yet to be investigated in a sample of New Zealand female adolescents from around the country. Methods: Adolescent females aged 15–18 y (*n* = 111) were recruited from 13 schools across eight locations throughout New Zealand to participate in this cross-sectional study. Participants completed questions about active commuting, and participation in organised sport and physical education, before wearing an Actigraph GT3X (Actigraph, Pensacola, FL, USA) +24 h a day for seven consecutive days to determine time spent in total, MVPA and light physical activity. Results: Active commuters accumulated 17 min/d (95% CI 8 to 26 min/d) more MVPA compared to those who did not. Those who participated in sport accumulated 45 min/d (95% CI 20 to 71 min/d) more light physical activity and 14 min/d (95% CI 5 to 23 min/d) more MVPA compared to those who did not. Participation in physical education did not seem to have a large impact on any component of physical activity. Participation in multiple domains of activity, e.g., active commuting and organised sport, was associated with higher accumulation of MVPA but not light activity. Conclusion Active commuting and sport both contribute a meaningful amount of daily MVPA. Sport participation has the potential to increase overall activity and displace sedentary behaviour. A combination of physical activity domains may be an important consideration when targeting ways to increase physical activity in adolescent females.

## 1. Introduction

The relationship between physical activity and positive health outcomes is well established. Higher levels of physical activity during adolescence, is associated with improved bone health [1] and weight status [2]. Additionally, physical actvity in adolescence has been shown to predict physical activity in adulthood [3,4] which is associated with improved cardiovascular health [5] and decreased risk of some cancers [6]. Furthermore, regular participation in physical activity can positively impact mental health [7,8]. 

To achieve these positive health outcomes, the recommendation for adolescents is to accumulate at least 60 min of moderate-to-vigorous physical activity (MVPA) per day [9]. However, globally in 2016, 81% of adolescents (85% of females and 78% of males) were not meeting country specific guidelines [10]. In New Zealand, the 2018 physical activity report card indicated that 39% of youth were meeting the physical activity guideline, with a 10% difference between males (46%) and females (36%) [11,12]. Traditionally, physical activity guidelines have focused on MVPA as it is associated with significant life-long health benefits [13]. However, based on increasing evidence to suggest that light physical activity also provides some health benefits [14], recommendations are increasingly including messages to replace sedentary time with light physical activity and/or MVPA [15,16]. 

Engagement in physical activity can occur across a number of domains, including leisure time, for example dancing or gardening in their free time, at school in the form of physical education or organised sport and for transport. There is good evidence to support active commuting as an important means of increasing physical activity in adolescents and whole populations [17,18]. More specifically, active commuting to and from school has been found to contribute 36% of daily MVPA [19]. However, a recent meta-analysis highlighted the gap in the literature regarding the impact of different domains of physical activity (including sports participation and physical education), on meeting guidelines [20].

Being enrolled in physical education at school has been shown to increase the likelihood of participants being classified as active particularly in younger children and adolescents [21]. However, many of the studies of adolescents assess physical activity by self-report which can provide inaccurate levels of habitual physical activity due to the introduction of systematic error such as social desirability, recall bias and inaccurate reporting by parental proxies [22,23]. Device measured physical activity is preferred over self-report as it provides better accuracy [24], less variability, more accurately determines sedentary time and activity and is not limited by recall bias [25]. Participants in physical education classes may be more likely to report time in that class as active, even when this is not always the case. There is also some evidence that indicates enrollment in physical education decreases with age [26,27]. Therefore, our current understanding of how participation in physical education classes’ impact device measured physical activity in older adolescent females is limited.

We know less about how participation in organised sport may impact physical activity, although intuitively a positive effect would be expected. However, participation declines sharply during the transition from childhood to adolescence [28,29], particularly in females [30]. These reasons may mean that participation in organised sport may not have a strong influence on the physical activity of older adolescent females.

Quantifying how active commuting, participation in organised sport and physical education contributes to daily physical activity either alone, or in combination has the potential to inform future interventions targeting physical activity in female adolescents. Therefore, this study aims to investigate the physical activity patterns of adolescent females in New Zealand in relation to active commuting, participation in organised sport and participation in school physical education both alone, and in combination.

## 2. Materials and Methods

### 2.1. Study Design 

The data used in this study were collected as part of the larger SuNDiAL project (Survey of Nutrition, Dietary Assessment and Lifestyle), the methods of which are described in detail elsewhere [31]. This project involved a cross sectional, nationwide survey of female adolescents. The main aim of SuNDiAL 2019 was to compare the dietary intakes of vegetarian and non-vegetarian adolescent females in New Zealand. A sample size of 300 high school female students enrolled from 13 high schools was calculated to give 80% power to the alpha = 0.05 level to detect a 0.5 standard deviation (SD) difference in any continuous outcome variables between vegetarians and non-vegetarians. This study was approved by the University of Otago Human Research Ethics Committee (H19/004). This trial was registered on the Australia, New Zealand Clinical trial website: ACTRN12619000290190.

### 2.2. Participants 

Two to five secondary schools in predetermined locations (based on the availability of data collectors) were contacted via email or phone and invited to participate if they had a female roll number of at least 200. Schools of a lower decile (a measure of socioeconomic status of the school) were preferentially invited in attempt to increase the diversity of the sample. In each location, when at least one school of the originally contacted schools did not agree to participate, other schools were contacted via email or word of mouth and invited to participate. Between February and September 2019, 13 secondary schools from eight different locations across New Zealand provided written consent and agreed to participate in the SuNDiAL project. Adolescent girls were then recruited from within each school via presentations. To be eligible to participate in the study, participants had to: self-identify as female, be between the age of 15 and 18 years (inclusive), be enrolled at one of the recruited secondary schools and speak and understand English. Participants were asked not to participate if they knew they were pregnant. Online informed consent was obtained from all participants, and the parent or guardian of participants under 16 years.

### 2.3. Outcome Measures 

#### 2.3.1. Demographics and Health 

Prior to the in-school data collection visit, participants completed an online questionnaire administered in REDCap (Research Electronic Data Capture (REDCap), production server version 9.3.3) (Vanderbilt University, Nashville, TN, USA). This questionnaire included basic demographic and health questions, but also asked participants “Do you take physical education as a subject at school”, “Do you play school sport”, “Do you play sport out of school” and “What is the most common way that you get to and from school?” to which they could answer one of the following: car, bus/train, bike, scooter/skateboard, walk or other (asked to specify). Active commuting was defined as scooter/skateboard, walking or biking.

#### 2.3.2. Socio-Economic Status

Household-level deprivation was assessed using the New Zealand Deprivation Index 2018 (NZDep2018) [32]. This used the home address of the participant to determine their index score based on census information about the neighbourhood area. Deciles of the NZDep2018 scores were then used to classify participants as living in low deprivation (deciles 1–3), medium deprivation (deciles 4–7), or high deprivation (deciles 8–10).

#### 2.3.3. Anthropometry

During school visits, weight (measured using one of Medisana PS420 (Nuess, Germany); Slater 9037 NK3R (Kent, United Kingdom); Seca Alpha 770 (Hamburg, Germany); or Soehnle Style Sense Comfort 400 scales (Backnang, Germany)) and height (measured using a Seca 213 (Hamburg, Germany) or Wedderburn (Sydney, Australia) stadiometer), were measured in duplicate to the nearest 0.1 kg or cm, respectively. Participants were measured wearing no shoes and in light clothing. A third measurement was taken when the difference between the first two measurements was more than 0.5 units and the mean of the two closest measurements was used as the true value. Body mass index (BMI) was calculated as weight (kg) divided by height (m), squared. BMI index z-scores were used to categorize participants as healthy weight, overweight or obese as per the WHO child growth standards [33].

#### 2.3.4. Accelerometry

Participants who consented to wear an accelerometer were provided with an Actigraph GT3X+ (Actigraph, Pensacola, FL, USA) which was worn on an elasticated belt over their right hip, continuously (24 h a day) for a seven-day period, with the exception of water-based activities including swimming, bathing, showering and during contact sport. Additionally, participants were asked to complete a sleep and wear time diary which involved recording; (1) times when the accelerometer was removed for more than five minutes, (2) activity that may not be recorded accurately, e.g., weight training, biking, swimming and the intensity of the activity and (3) sleep data, including: the time the participants tried to go to sleep and the time they woke up and got out of bed. The accelerometers were initialised at a sampling rate of 30 Hz. Accelerometers were provided to participants on a first in, first served basis. At certain times, in some locations, not every participant who consented to accelerometry was provided with an accelerometer due to the finite number of devices available. Accelerometers and wear time diaries were collected at the end of the seven-day period.

### 2.4. Data Processing 

#### 2.4.1. Wear Time 

Non wear time was defined as when the participants reported accelerometer removal in their wear time diary. Days were considered valid if wear time during waking hours was 10 h or more, total wear time or wear time plus imputed physical activity (activity reported in the wear time diary, e.g., swimming) equalled 20 h or more and included at least 2 h of sleep. Data were included if three or more valid days were available (as this is the most commonly used requirement) [34].

#### 2.4.2. ActiGraph Data 

Time stamped data from the ActiGraph accelerometers were downloaded using Actilife software, saved in 60 s epochs in .csv format and imported into Stata (Version 14.1 for Mac; StataCorp, College Station, TX, USA). Sedentary time was defined as the amount of time counts did not exceed 150 counts/min using the y-axis [35]. Light physical activity and MVPA were therefore defined as 150 to 1951 counts/min and at least 1952 counts/min, respectively, using the y-axis [36]. The Sadeh algorithm [37], was used to identify time spent asleep between self-reported attempted sleep time and wake up times. The Sadeh algorithm is validated for wrist worn accelerometery and has also been shown to perform well with hip worn accelerometer data particularly when it is constrained by self-reported bedtimes [38]. Where physical activity was completed when the participant was not wearing the accelerometer (e.g. swimming) it was categorised as ‘non-wear time physical activity’. Non-wear time physical activity was added to the amount of light physical activity and MVPA using participant reported intensity, to calculate total physical activity, light physical activity and MVPA. 

#### 2.4.3. Hour before and after School

To identify activity in the hour directly before and after school, individual school start and end times were obtained via secondary school websites. Where these times were not published online, they were obtained via phone contact with the school administration.

### 2.5. Statistical Analysis 

All statistical analysis was carried out in Stata 16.1 (StataCorp, College Station, TX, USA). T-tests and Fisher’s exact tests were used to assess demographic differences between those with and without accelerometry data, those who did and did not actively commute, those who took physical education at school and those who did not, and those who did and did not play sport. 

To estimate the mean differences (95% CI) in physical activity between domains (e.g., between those who actively commuted compared to those who did not), mixed effects regression models were used. School was included as a random effect and estimates were adjusted for age, deprivation, and BMI z-score. A further, fully adjusted model included all domains: active commuting, sport, and physical education. Residuals of models were plotted and visually assessed for homoskedasticity and normality. Collinearity was assessed by variance inflation factors for the model that included all domains (all VIF were less than 1.43).

To determine the mean difference (95% CI) in physical activity in the hour before and after school between those who actively commuted and those who did not, mixed effects regression models were used with the number of minutes in light physical activity, MVPA, and total physical activity during that hour as the dependent variable and active commuter or not as the independent variable. School was included as a random effect. As sport can sometimes take place in the hour before or after school, models were also run adjusted for whether or not the participant played sport. 

To illustrate the physical activity differences for different combinations of domains (active commuting, organised sport and physical education), box plots were generated, which give a transparent summary of all available data and indicate summary variables (i.e., minimum, maximum, median, and 25th and 75th percentiles).

## 3. Results

### 3.1. Participant Characteristics

A total of 272 participants participated in at least one aspect of the SuNDiAL project. Of those, 111 participants provided valid accelerometry data (Figure 1). Participants who provided valid accelerometry data were similar in terms of age, deprivation, and BMI z-score (Table 1). The mean age of participants who provided valid accelerometry data was 16.8 years (SD 0.9) and 65% of the sample were a healthy weight.

Just under one quarter of the participants who provided accelerometer data commuted actively to school, most (almost 90%) reported walking, while the remainder reported commuting via bike. More than one third participated in physical education (36%) at school, while 77 participants (69%) reported participating in sport either at school (*n* = 21), outside of school (*n* = 12), or both (*n* = 44). Participants who reported playing sport were more likely to also be taking physical education classes at school, and were less likely to be from an area of high deprivation (Table 1).

### 3.2. Physical Activity Patterns 

Overall 24 h activity patterns of this sample have been reported elsewhere [39]. Mean ActiGraph wear time was 1410 min/d (23 h 30 min), for all participants. Average time per day spent in light physical activity, was 226 min (2 h 46 min), while time spent it MVPA was 44 min. Overall, 24% (27 out of 111) of participants met the physical activity guidelines of at least 60 min of MVPA per day. 

### 3.3. Active Commuting, Physical Education and Organised Sport as Predictors of Physical Activity 

Compared to the group who did not actively commute; those who did accumulated 16 min (95% CI 8 to 25 min) more MVPA per day, while differences in light and total physical activity were not statistically significant (Table 2). However, when looking specifically at when the time commuting was likely to occur (in the hour before and after school), and after adjusting for sport participation (which could also occur in this time frame), active commuters accumulated 4.2 min (95% CI 1.1 to 7.4 min) more light activity in the hour before school, and 5.3 min (95% CI 3.3 to 7.3 min) and 4.6 min (95% CI 2.4 to 6.8 min) more MVPA in the hour before and after school, respectively (Table 3). 

Participating in physical education classes at school did not seem to influence MVPA, but small differences in light and total physical activity were indicated (24 min (95% CI −3 to 50 min) and 28 min (95% CI −1 to 58) min, respectively) (Table 2). 

Compared to those who did not participate in organised sport, those who did accumulated 38 min (95% CI 12 to 65 min) more light physical activity, 13 min (95% CI 5 to 22 min) more MVPA and 51 min (95% CI 20 to 81 min) more total physical activity. 

Box plots of physical activity stratified by domains, and combinations of domains illustrate that activity levels vary substantially within each domain (Figure 2). Those people who did none of the domains of interest had the lowest median (IQR) level of MVPA (29.0 min (21.7 to 40.3 min)). Participating in one, two or three domains appeared to have an additive effect for some combinations, so that those who reported participation in all three domains had the highest median MVPA, with more than half this group recording an average of 69 min (56.5 to 75.8 min) of MVPA across the wear period–the exception being the combination of active commuting and physical education–although only one participant reported this combination. Median levels of light physical activity did not differ markedly across the different combinations of domains except in participants who did all three domains, where the median was higher. 

## 4. Discussion

The results from this study indicate that when compared to others who do not actively commute, or participate in organised sport, those who do, accumulate nearly a quarter of an hour more daily MVPA. Interestingly, those who participated in organised sport also accumulated a greater amount of light and total physical activity. In contrast to other studies, we found that participation in physical education classes had little impact on physical activity levels in this sample of female adolescents. 

This study is the first to report the effects of organised sport participation on light physical activity in NZ adolescent females. Participation in organised sport was associated with the accumulation of greater than 30 min more light physical activity, in addition to higher rates of MVPA. This result indicates that those who participated in organised sport had a more positive overall activity pattern than those who did not [20]. As light physical activity has received far less attention in research to date, this novel finding highlights the potential that participation in organised sport has to influence not just MVPA, but other components of the day that may also positively affect health [14].

We must acknowledge, however, that the impact of participation in organised sport on physical activity we observed is almost twice as large as that observed by comparable studies of Australian adolescents [40,41]. Differences in the representativeness of the samples, combined with methodological differences in the assessment of physical activity make further comparisons between these studies difficult. However, we believe that using organised sport as a means of increasing both MVPA and total physical activity in female adolescents’ warrants further investigation. 

The fact that those who actively commuted accumulated more MVPA than those who did not was not unexpected as there is a large pool of evidence which suggests that active commuters accumulate more MVPA [18,42,43,44]. In the present study, participants who actively commuted accumulated a combined total of 9 min/d more MVPA in the hour directly before and after school. This finding was more than twice that reported in a sample of adolescent females from the USA [42]. Interestingly, 33% of active commuters’ MVPA was accumulated in the hour before and after school, which more than accounts for the difference in MVPA when compared to those who did not actively commute, confirming that it is highly likely that the commuting is responsible for the difference in MVPA [43]. However, while those who actively commuted accumulated more MVPA than their peers who used motorised transport to get to school, their light and total activity was not notably different. This finding may suggest compensatory behaviour by those who actively commute. Compensatory behaviour has also been observed in a study of Australian adolescents who for every additional 10 min of MVPA, accumulated 25 min less LPA and 5 min less MVPA the following day [45]. The findings from this study contribute to the pool of evidence which supports the promotion of active commuting as a means to increase MVPA in this age group, however possible compensatory behaviors means that the impact of active commuting on 24 h activity should be investigated further.

The New Zealand physical education curriculum includes seven main areas of study including mental health, sexual education, food and nutrition, body care and physical safety, physical activity, sports studies and outdoor education [46]; only two of which are likely to include components of physical activity. A recent meta-analysis highlighted that just over one third of time spent in physical education class included MVPA and the time spent in MVPA during these classes decreases further in higher-income countries [21]. These factors may, at least in part, explain why enrolment in physical education classes did not appear to have a meaningful impact on MVPA, light or total physical activity in this study. Furthermore, physical education classes may only be scheduled 2–4 times a week, limiting the number of classes that would have been attended during accelerometer wear. Future research would benefit from more targeted analysis of the physical activity component of physical education classes in New Zealand female adolescents to identify the amount of physical activity included in the curriculum, frequency of classes, and the impact this has on overall physical activity. 

Another seemingly unique aspect of this study was the analysis of how different domains of activity combined to impact on both MVPA and light activity. While the small numbers in each category limit the conclusions we can draw from this analysis it is interesting to note that the majority of female adolescents who do physical education as a subject at school also participate in organised sport, and that the combination of organised sport and active commuting seem to result in the highest MVPA accumulation (across the 2 domain combinations). It is possible that the greatest increases in physical activity could occur from an intervention designed to target multiple domains in which activity can be accumulated. This suggestion aligns with the recently released WHO guidelines on physical activity and sedentary behaviour which emphasise that any amount or type of activity can contribute to overall health [9].

### Strengths and Limitations

By using device measured physical activity, issues associated with self-reported data are minimised. Furthermore, a unique component of this study is the assessment of light physical activity, which is often not reported in physical activity research in adolescents. This study included a wide spread of female adolescents from 13 schools across eight locations in New Zealand, which resulted in a more heterogeneous sample than other studies conducted here. This study does however have a number of limitations. Firstly, the sample was a medium sized convenience sample that despite being nationwide, may limit the generalizability of the findings to the wider New Zealand population. Participants who consented to the study also may have been more likely to participate in organised sport and physical education compared to the general population as an interest in health/activity may have sparked interest in participation. We chose to use the Freedson cut points for accelerometer data as there are few cut points which have been validated for the age group included in this study (the upper range for children/adolescent validations is 16 years) [36]; furthermore, the physical size of the participants (aged 15 to 18 years), is likely to be more similar to adults compared to younger adolescents. There is also considerable variability among studies using accelerometry with regard to epoch length, non-wear time definitions, wear time criteria cut point and valid wear time for inclusion [34]. However, comparisons were made within one data set, and while the use of different cut points and epoch lengths, etc. may have influenced the absolute values reported here, it seems unlikely they would have had a meaningful impact on the associations reported.

## 5. Conclusions

Those who actively commuted or participated in sport accumulated the highest levels of MVPA, while physical education classes had little impact on MVPA in this group of female adolescents. Participation in sport appeared to have the added benefit of higher light and total activity levels in addition to higher MVPA. Programs that target active commuting and organised sport, or better, programs that include a combination of physical activity domains, have the potential to improve both the MVPA and total activity of female adolescents and therefore, may result in positive health outcomes throughout adolescence which may continue into adulthood.

## Figures and Tables

**Figure 1 ijerph-18-08077-f001:**
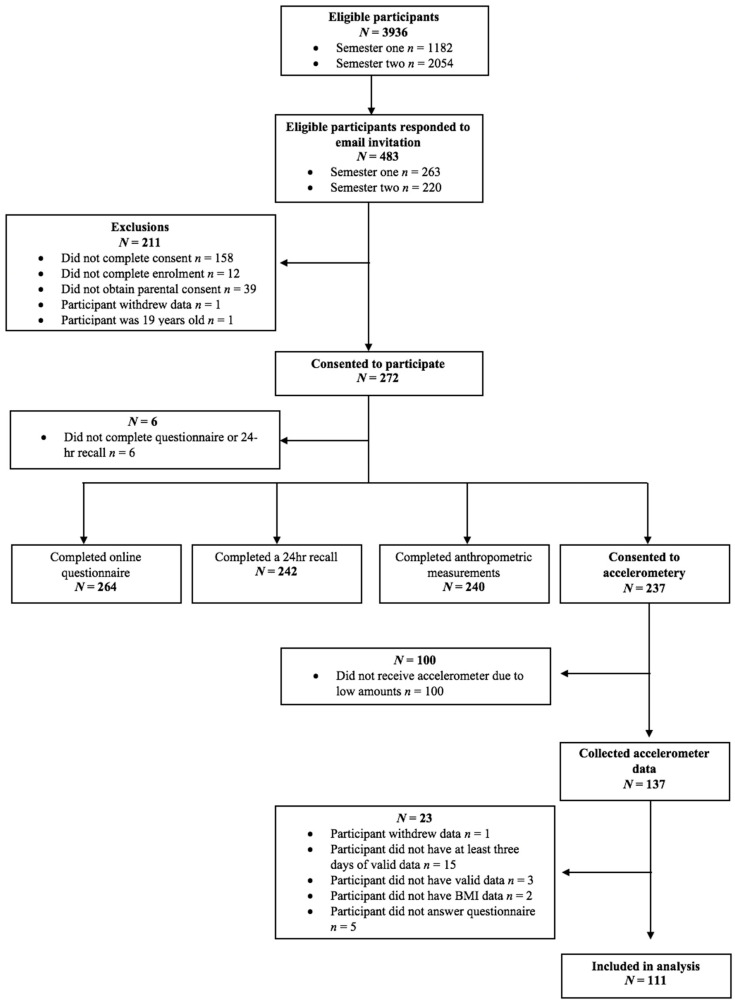
SuNDiAL Study Participant Flow Chart.

**Figure 2 ijerph-18-08077-f002:**
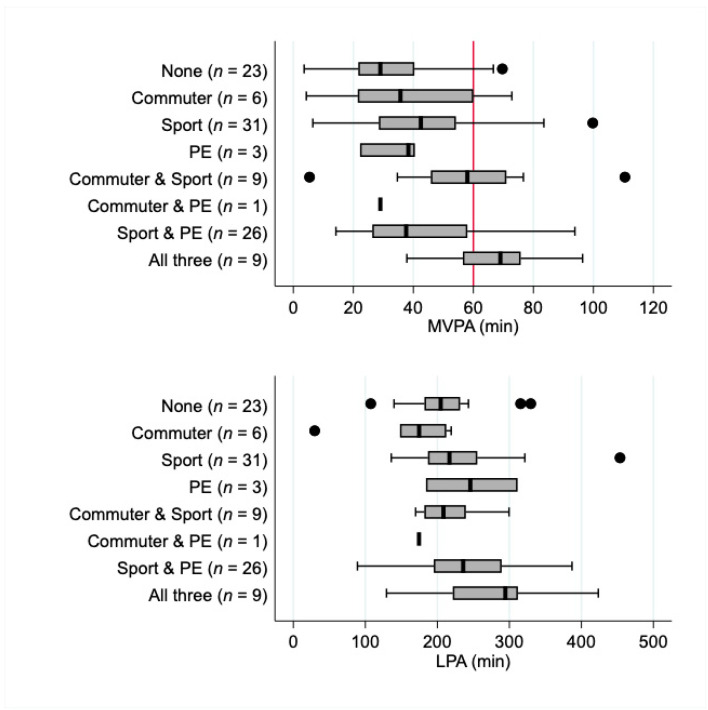
Box plots of light physical activity and MVPA stratified by combinations of domains of physical activity. Numbers of participants in each combination group are indicated. The dots represent ‘outliers’ who had values that were greater than 1.5 times the IQR away from the upper or lower quartile.

**Table 1 ijerph-18-08077-t001:** Characteristics of sample.

	Full Sample	Accelerometry Sample	Active Commuters	Participants Who Do Physical Education Classes at School	Participants Who Play Sport
N	272	111	26	40	77
Age, mean (SD) y	16.8 (0.8)	16.8 (0.9)	16.7 (0.9)	16.5 (0.9) ^e^	16.8 (0.8)
Household level deprivation ^a^, *n* (%)					
Low	109 (40.4)	39 (35.1)	8 (30.8)	21 (52.5) ^e^	29 (37.7)
Medium	107 (39.6)	49 (44.1)	13 (50.0)	14 (35.0)	39 (50.7)
High	54 (20.0)	23 (20.7)	5 (19.2)	5 (12.5)	9 (11.7) ^e^
BMI z-score ^b^, mean (SD)	0.74 (0.96)	0.72 (0.89)	0.64 (1.03)	0.70 (0.94)	0.68 (0.83)
Weight status ^b^, *n* (%)					
Healthy	157 (65.4)	72 (64.9)	15 (57.7)	26 (65.0)	51 (66.2)
Overweight	57 (23.8)	31 (27.9)	9 (34.6)	11 (27.5)	22 (28.6)
Obese	26 (10.8)	8 (7.2)	2 (7.7)	3 (7.5)	4 (5.2)
School commute ^c^, *n* (%)					
Car	120 (48.8)	55 (49.6)	0	17 (42.5)	40 (52.0)
Bus or Train	65 (26.4)	30 (27.0)	0	12 (30.0)	18 (23.4)
Bike	8 (3.3)	3 (2.7)	3 (11.5)	2 (5.0)	3 (3.9)
Walk	53 (21.5)	23 (20.7)	23 (88.5)	9 (22.5)	16 (20.8)
Physical education classes, *n* (%)	93 (37.8)	40 (36.0)	11 (42.3)	40 (100)	36 (46.8) ^e^
Play sport, *n* (%)					
No Sport	89 (36.2)	34 (30.6)	7 (26.9)	4 (10.0) ^e^	0
School Sport	46 (18.7)	21 (18.9)	3 (11.5)	12 (30.0)	21 (27.3)
Outside of School sport	28 (11.4)	12 (10.8)	6 (23.1)	3 (7.5)	12 (15.6)
Both	83 (33.7)	44 (39.6)	10 (38.5)	21 (52.5)	45 (57.7)
LPA ^d^, mean (SD) min	-	226 (66)	227 (81)	250 (71) ^e^	238 (67) ^e^
MVPA ^d^, mean (SD) min	-	43.6 (22.8)	56.4 (25.7) ^e^	48.4 (23.0)	48.3 (23.4) ^e^
Total physical activity ^d^, mean (SD) min		270 (75)	283 (75)	298 (75) ^e^	286 (73) ^e^


^a^ Measured using home address applied to the New Zealand Deprivation Index (2018): Low is deciles 1–3; Medium is deciles 4–7; high is deciles 8–10. Two participants in the full sample were missing home address information. ^b^ BMI z-scores calculated using WHO growth charts: Healthy is BMI z-score < 1; Overweight is BMI z-score ≥ 1 & <2; Obese is BMI z-score ≥ 2. Thirty-two participants in the full sample were missing BMI z-score data. ^c^ Active commuters are defined as walking or biking some part of their commute to school (no participants indicated that they skated or scootered to school). Four participants indicated that they did a mixture of motorised and active commuting (assigned to active). Twenty-six participants in the full sample did not complete the questionnaire with the commuting, PE, and sport questions. ^d^ LPA: Light physical activity; MVPA: Moderate-to-Vigorous physical activity; Total physical activity: sum of LPA and MVPA; measured as mean minutes per day from accelerometers. ^e^ Significantly different to those not in this group, *p* < 0.05.

**Table 2 ijerph-18-08077-t002:** Active commuting, physical education classes, and sport as predictors of physical activity (*n* = 111).

	Mean Difference (95% CI) ^a^, Min	*p*-Value	Adjusted Mean Difference (95% CI) ^a^, Min	*p*-Value
Light Physical Activity				
Active commuter	1 (−27, 29)	0.958	−3 (−30, 23)	0.814
Physical education classes	36 (10, 62)	0.007	24 (−3, 50)	0.082
Sport	45 (20, 71)	0.001	38 (12, 65)	0.005
Moderate-to-vigorous physical activity				
Active commuter	17 (8, 26)	<0.001	16 (8, 25)	<0.001
Physical education classes	6 (−3, 15)	0.186	1 (−8, 9)	0.908
Sport	14 (5, 23)	0.002	13 (5, 22)	0.003
Total physical activity				
Active commuter	19 (−13, 51)	0.248	15 (−15, 45)	0.320
Physical education classes	46 (17, 75)	0.002	28 (−1, 58)	0.062
Sport	60 (31, 89)	<0.001	51 (20, 81)	0.001

^a^ Mean differences (95% CI) and *p*-values for physical activity minutes were estimated using a mixed effect regression model, with school as a random effect and adjusted for age, NZDep, and BMI z-score. The adjusted mean differences are estimated after adjustment for other predictors reported in the table.

**Table 3 ijerph-18-08077-t003:** Physical activity in the hour before and the hour after school in active and non-active commuters (*n* = 108).

	Not Active Commuters, Mean (SD) Min (*n* = 83)	Active Commuters, Mean (SD) Min (*n* = 25)	Mean Difference ^a^ (95% CI) Min	*p*-Value	Mean Difference ^a^ (95% CI) Min Adjusted for Sport Participation	*p*-Value
Light physical activity						
In the hour before school	17.0 (7.5)	20.9 (9.7)	4.5 (1.2, 7.7)	0.007	4.2 (1.1, 7.4)	0.009
In the hour after school	16.0 (6.4)	17.2 (5.8)	1.4 (−1.4, 4.1)	0.324	1.3 (−1.4, 4.1)	0.337
Moderate-to-vigorous physical activity						
In the hour before school	2.9 (2.5)	8.1 (8.5)	5.3 (3.3, 7.3)	<0.001	5.3 (3.3, 7.3)	<0.001
In the hour after school	5.8 (4.2)	10.3 (7.3)	4.6 (2.3, 6.8)	<0.001	4.6 (2.4, 6.8)	<0.001
Total physical activity ^b^						
In the hour before school	19.8 (7.5)	29.1 (10.9)	9.6 (6.0, 13.2)	<0.001	9.3 (5.8, 12.8)	<0.001
In the hour after school	21.8 (8.3)	27.4 (10.7)	5.8 (1.9, 9.7)	0.004	5.8 (1.8, 9.7)	0.004

^a^ Mean differences (95% CI) and *p*-values estimated using a mixed effects regression model with school as a random effect. ^b^ Total physical activity is light physical activity plus moderate-to-vigorous physical activity.

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
