# Peer review of "The Impact of Organised Sport, Physical Education and Active Commuting on Physical Activity in a Sample of New Zealand Adolescent Females"

_ijerph, 2021, doi:10.3390/ijerph18158077_

Round 1

Reviewer 1 Report

This study assesses different domains of physical activity in adolescent girls in New Zealand using both subjective and device-based measures. It is a well-constructed study, albeit under-powered in terms of sample size. However, there are a few updates and clarifications which need to be made before this study can be considered for publication.

General comment

One of the things which concerns me about this study is the small sample size. You are quite a bit below your sample size calculation quoted in Lines 81-83. Having a quick look on PubMed, I quickly found studies which used accelerometry in adolescents but had much higher sample sizes (see below for a couple of examples):

Hamer M, Patalay P, Bell S, Batty GD. Change in device-measured physical activity assessed in childhood and adolescence in relation to depressive symptoms: a general population-based cohort study. J Epidemiol Community Health. 2020 Apr 1;74(4):330-5.

Kandola A, Lewis G, Osborn DP, Stubbs B, Hayes JF. Device-measured sedentary behaviour and anxiety symptoms during adolescence: a 6-year prospective cohort study. Psychological Medicine. 2020 Dec 18:1-0.

I guess my query is whether this study is suitably powered enough to give any weight to your findings?

Introduction

Lines 35-37: As the study focuses on adolescents only, it is important to refer to relevant positive health outcomes (i.e. reduced cardiovascular disease and cancer are not necessarily the most relevant benefits for this age group).

Lines 56-63: This paragraph could be stronger. For example, you need to more clearly state why assessing by self-report is an issue (e.g. self-report bias, social desirability bias) and give a reference for this. Also, there needs to be a better lead-in to the fact that device-measured physical activity could be a way of addressing this issue.

Materials and Methods

Line 124: Update to “GT3X+”.

Line 146: Clarify the Hz which data was collected in? Guessing 30Hz?

Lines 147-150: I have concerns about the cut points you have chosen. These are usually used in adults (i.e. see the chosen references), not in children or adolescents. Also, a mixture of vector magnitude and vertical cut points have also been used which seems strange… Please clarify these decisions. Why did you not use some of the established children’s cut points?

Line 150: Apologies for my ignorance, but if the Sadeh algorithm has been used to determine sleep times, why was the sleep diary also needed?

Line 155: Fully highlight “PA” as “physical activity”.

Line 159: It would be good to make it clear how these times were incorporated into Stata.

Line 176: What maximum allowable value was used for the VIF?

Results

Line 203: Replace “where” with “were”.

Discussion

Line 256: Update to “more daily MVPA”.

Line 273-276: I would include a reference in this section highlighting studies which have shown the beneficial impacts of light physical activity.

Conclusion

Lines 340-341: You could allude a little further on the finding regarding physical education classes. What do you suggest needs to be done to enhance physical education’s impact on MVPA in adolescent females?

Reviewer 2 Report

The article presents a topic as important as the practice of physical activity in adolescents in a sample of the New Zealand population. Although it is not very original, given that the conclusions reached by the authors have already been addressed in other studies, it may be one more contribution to scientific knowledge.

I congratulate the authors for their work and I thank the review of this manuscript, which is within my line of work.

After passing the plagiarism detector, the percentage is 39%, 25% excluding sources. Some snippets are common expressions so it would be within the allowed margins.

Next I go on to make a series of suggestions for authors that I think will further improve the quality of their work.

Abstract
The authors should indicate in the first line which entity recommends these optimal levels of physical activity in adolescents.

 Introduction
On line 49, when talking about different scenarios where physical activity can be done, the authors must add more scenarios and write some more substantiation on each one of them:
- Physical activity in free time.
- Federated Physical Activity.
- Physical Activity in Physical Education.
- Physical activity in active breaks.
- Physical Activity in active transport.
- Physical activity at home.

 On line 64, I suggest incorporating the data from this recent study, which also finds low levels of practice as the children get older. https://www.mdpi.com/2071-1050/13/14/7806 

Method
The authors present in an orderly and clear way the information related to the sample, protocol, as well as data analysis.
The authors correctly incorporate the ethical aspects.
Indicate the method followed for the selection of participating schools.
Describe what is understood by the concept of organized sport (adolescent practicing more than 4 days a week, adolescent with participation in federated competitions, adolescent with federated license participating or not in competitions, etc.) or how this variable was reflected in the questionnaire.

Results
This section is correct, the results are well presented and are consistent with the objectives and hypotheses of the study.

Discussion
 Although the discussion is correct, I consider that a contribution of more references to similar studies would further enrich the quality of the work carried out by the authors.

It would be recommended that the authors contribute their point of view on how to improve the levels of physical activity practice in adolescents, at the end of their work as a strategy to be carried out.

Round 2

Reviewer 1 Report

Thank you for addressing my comments. I now feel this paper is ready to be published.